# Assessment of Bacterial Nanocellulose Loaded with Acetylsalicylic Acid or Povidone-Iodine as Bioactive Dressings for Skin and Soft Tissue Infections

**DOI:** 10.3390/pharmaceutics14081661

**Published:** 2022-08-09

**Authors:** Shaydier Argel, Melissa Castaño, Daiver Estiven Jimenez, Sebastian Rodríguez, Maria Jose Vallejo, Cristina Isabel Castro, Marlon Andres Osorio

**Affiliations:** 1Nanotechnology Engineering Program, School of Engineering, Universidad Pontificia Bolivariana, Circular 1 #70-01, Medellin 050031, Colombia; 2New Materials Research Group, Universidad Pontificia Bolivariana, Circular 1 #70-01, Medellin 050031, Colombia; 3Biology Systems Research Group, School of Health Science, Universidad Pontificia Bolivariana, Cl. 78b #72a-159, Medellin 050034, Colombia

**Keywords:** bacterial nanocellulose, wound dressing, povidone-iodine, acetylsalicylic acid

## Abstract

Bacterial nanocellulose (BNC) is a novel nanomaterial known for its large surface area, biocompatibility, and non-toxicity. BNC contributes to regenerative processes in the skin but lacks antimicrobial and anti-inflammatory properties. Herein, the development of bioactive wound dressings by loading antibacterial povidone-iodine (PVI) or anti-inflammatory acetylsalicylic acid (ASA) into bacterial cellulose is presented. BNC is produced using Hestrin–Schramm culture media and loaded via immersion in PVI and ASA. Through scanning electron microscopy, BNC reveals open porosity where the bioactive compounds are loaded; the mechanical tests show that the dressing prevents mechanical wear. The loading kinetic and release assays (using the Franz cell method) under simulated fluids present a maximum loading of 589.36 mg PVI/g BNC and 38.61 mg ASA/g BNC, and both systems present a slow release profile at 24 h. Through histology, the complete diffusion of the bioactive compounds is observed across the layers of porcine skin. Finally, in the antimicrobial experiment, BNC/PVI produced an inhibition halo for Gram-positive and Gram-negative bacteria, confirming the antibacterial activity. Meanwhile, the protein denaturation test shows effective anti-inflammatory activity in BNC/ASA dressings. Accordingly, BNC is a suitable platform for the development of bioactive wound dressings, particularly those with antibacterial and anti-inflammatory properties.

## 1. Introduction

The wound healing process prevents bacterial contamination and recovers the homeostasis of the body once skin structures are compromised due to external damage [1]. However, when bacterial contamination occurs, patients can present skin and soft tissue infections (SSTIs), a type of condition that affects around 14 million people every year in the United States [2]. SSTIs promote the deterioration of granulation tissue, collagen, elastin, fibrin, and growth factors (causing inflammation and pain), affecting the quality of life of the patients [3]. Therefore, it is crucial to pursue the development of bioactive wound dressings in order to maintain a physiological moisture environment, provide wound oxygenation, and set up biological signals to promote healing processes, such as antimicrobials or anti-inflammatories [4]. Dressings have been developed using hydrogels, hydrocolloids, and biopolymers to play an essential role as matrices for delivery systems that fulfill specific functions in wounds [5].

Bacterial nanocellulose (BNC) is a natural, non-toxic, biocompatible hydrogel that presents high purity and water absorption capacity, as well as good mechanical properties and biomimicry of the soft tissue extracellular matrix [6]. Most of these properties originate from BNC’s unique tridimensional nanofibrillar network, which contains fibers 100 times thinner than conventional cellulose (cotton) dressings [7]. In this way, BNC acts as a mechanical barrier to microbial contamination and external damage, but also represents a suitable carrier for the inclusion of drugs and therefore for the development of controlled release systems [8]. Moreover, BNC has been proven to be a promising biomaterial that fulfills all performance conditions for an ideal wound dressing [9]. However, BNC has no antimicrobial, analgesic, or anti-inflammatory properties [7], properties that nowadays are crucial for the treatment of chronic and infected wounds. For instance, the presence of microorganisms generates infections and inflammation, increasing the healing time by disrupting the re-epithelialization process and subsequently causing pain [10,11]. Therefore, compounds such as antibiotics and pain killers could complement treatments with wound dressing, making bacterial nanocellulose a suitable material for interactive wound dressings.

BNC wound dressings have been reported in the literature. For instance, Berndt et al. (2013) report the development of a porous hybrid consisting of bacterial nanocellulose and silver nanoparticles for the development of antimicrobial dressings; in addition, silver ions have been used in order to achieve the same aim; however, the excessive use of these ions has been reported, which can cause adverse effects on human cells [12,13]. Wiegand et al. (2015) found that the implementation of polyhexanide in bacterial nanocellulose from *Komagataeibacter xylinus* presents successful antibacterial properties. However, polyhexanide’s clinical efficacy and safety are poorly documented, as it is formulated with a buffer and without other constituents, such as humectants or moisturizers [14,15]. Similarly, the use of other drugs, such as analgesics, anesthetics, beta-blockers, and antibiotics, has been described for the preparation of drug delivery systems with bacterial nanocellulose [7,9,16,17,18,19]. Accordingly, there is broad potential for the investigation of novel substances with anti-inflammatory or antibacterial properties to be loaded on BNC.

Among these potential compounds, povidone-iodine (PVI) and acetylsalicylic acid (ASA) can be highlighted. PVI is a water-soluble complex widely used as a skin disinfectant due to its prolonged germicidal action against Gram-positive and Gram-negative bacteria, as well as viruses, fungi, and microbes, whose cell walls develop pores as a result of PVI´s irreversible binding to their proteins, lipids, and nucleic acids, which leads to cell membrane damage [15,20,21]. On the other hand, ASA is an analgesic and nonsteroidal anti-inflammatory agent [22,23]; thus, ASA can increase anti-inflammatory molecules such as lipoxins, which possess pro-repair and pro-resolution qualities, and reduce excessive tissue injury and chronic inflammation [24]. Furthermore, PVI is one of the most widely used antiseptics for skin [25], and it has been used in wound dressings, specifically in woven and non-woven dressings [5], alginate films [26], and even in BNC, where it showed the highest antibiofilm activity among several commonly used antiseptics [21].

Therefore, the aim of this work is the development of bacterial nanocellulose wound dressings from the strain *Komagateibacter medellinensis*, specifically for the treatment of SSTI wounds, in which the healing process is aided by anti-inflammatory, antimicrobial, and analgesic bioactive compounds. Moreover, this study characterizes the loading and release dynamics, as well as the mechanical and morphological properties, of two types of formulations: one consisting of PVI due to its antimicrobial activity and another one with ASA due to its anti-inflammatory and analgesic properties.

## 2. Materials and Methods

### 2.1. Materials

Peptone, yeast extract, glucose, anhydrous sodium dihydrogen phosphate, magnesium sulfate, and potassium dihydrogen phosphate were all acquired from Sigma Aldrich, and citric acid from AppliChem and acetic acid from Merck. For bioactive wound dressings, acetylsalicylic acid was purchased from Sigma Aldrich and povidone-iodine from Lodiger. In the complexation process, sodium hydroxide and potassium iodate were purchased from Merck, ferric chloride from Sigma, and starch from Fisher Scientific. To simulate acidic sweat, sodium chloride and L-histidine were purchased from Merck and lactic acid from Sigma Aldrich.

### 2.2. Development of Bioactive Wound Dressings

BNC was produced using Hestrin–Schramm (HS) culture medium, described in the literature [27]. The medium was adjusted to pH = 3.6 with citric acid and then autoclaved at 15 psi for 15 min. Then, 45 mL of culture medium was inoculated with *Komagateibacter medellinensis* at 0.5 McFarland. Fermentation was carried out under static conditions for 7 days at 28 °C. Dressings were purified in 5 wt. % KOH solution at room temperature for 14 h, followed by continuous rising with water until a neutral pH was reached.

### 2.3. Preparation of BNC Wound Dressings

Bioactive compounds were introduced into BNC dressings by immersion in 10 wt. % PVI and 10 wt. % ASA topical formulations for 3 h, respectively, at room temperature. Once the dressings were loaded, they were removed from solutions and stored until use. Dressings were named BNC/PVI and BNC/ASA. The concentration values of PVI were chosen based on data reported in the literature [28,29,30]. Concentrations of 10 wt. % commonly do not inhibit the granulation and epithelialization processes and wounds have shown faster neovascularization with PVI treatment compared with controls [28]. Likewise, topical applications of ASA at 10 wt. % promote the granulation and epithelialization processes and provide the desired therapeutic effect [29,30].

### 2.4. Wound Dressing Characterization

Scanning electron microscopy (SEM)SEM was used to analyze the morphology of BNC dressings. The dressings were frozen at −196 °C using liquid nitrogen and freeze-dried for 24 h under 0.020 mBar of vacuum; then, the samples were coated with gold using ion sputtering. Finally, the samples were observed at 5000 and 20,000 magnifications using a Jeol JSM 5910 LV Scanning Electron Microscope operating at 20 kV.Mechanical characterizationMechanical properties of BNC/PVI and BNC/ASA were analyzed using uniaxial tensile tests in an Instron Universal Testing Machine. The specimens were cut following ASTM D882 guidelines, producing samples with a thickness of 0.55 mm and length of 20 mm; samples were mounted in pneumatic clamps and stretched up to the point of failure. Twelve curves each were obtained for PVI and ASA; the representative curve was the average of all measurements. Young’s modulus, strain at break, and stress at break were obtained.Loading and release experimentsBNC dressings were deposited separately in 100 mL of both formulations (PVI and ASA) for 1 h without agitation, and aliquots of 100 µL were taken at 5, 10, 15, 20, 25, 30, 35, 40, 50, and 60 min. To measure the concentrations of PVI and ASA, a complexation process was performed. For PVI, the protocol established by Sulistyarti et al. (2015) was followed: 50 µL of the aliquot was diluted in 5 mL of KIO ${}_{3}$ 3 mM; then, 100 µL was taken from the earlier solution and mixed with 100 µL of starch at 0.5 wt. %. The concentrations were determined using UV–Vis spectroscopy at a wavelength of 615 nm [31]. Following the protocol established by Chambers et al. (1993) for ASA, 100 µL of the aliquot was diluted in 8.9 mL of water and 1 mL of NaOH 2M; then, 100 µL from the earlier solution was diluted with FeCl ${}_{3}$ 0.02 M. Then, the concentrations were measured with UV–Vis spectroscopy with a wavelength of 530 nm [32].The experimental data were modeled using kinetics for the PVI and ASA loaded (active compounds) into the adsorbent (BNC). The experimental loading capacity *Q*_*t*_ of active compounds was calculated by the following equation [33,34]:
(1)$$Q_{t}=\frac{(C_{0}-C_{t})\cdot V}{W}$$
where $C_{0}$ is the initial concentration of the compound in the solution (g/L); $C_{t}$ is the concentration in an instant *t* (g/L); *V* is the volume of the solution, which was constant throughout the assay (L); *W* is the weight of the BNC dressing (g). Then, $Q_{t}$ is the mass absorbed by compounds by the mass of BNC (g of active compounds/g of BNC).Lagergren’s pseudo-first-order model was used as an loading model [34,35]. The model is described mathematically by the following equation:
(2)$$\frac{dQ_{t}}{dt}=K\left(Q_{e}-Q_{t}\right)$$
where *K* is the first-order kinetic constant. This constant represents the velocity of the process and can give an indication of the affinity between the components evaluated in the loading experiment, and $Q_{e}$ is the concentration at equilibrium. Considering the adsorbent free of solute, initially, the resulting equation obtained in an integrated manner is
(3)$$Q_{t}=Q_{e}\left(1-e^{-K_{1}t}\right)$$For release experiments, BNC/PVI and BNC/ASA dressings were initially oven-dried at 37 ºC for 1 h to remove excess liquid, and then two release kinetic assays were performed using static diffusion cells, following the method of Franz et al. [20]. Figure 1 shows a schematic image representing the assay performed. In the first test, the receptor compartment had an average volume of 12 mL and a donor compartment of 3 mL and was kept at 37 °C using a heating plate. Water and simulated acidic sweat media were used to reproduce the physiological working conditions of the dressings; acidic sweat was prepared following the AATCC 125-2004 [36]. The solution in each cell was continuously stirred using a magnetic stirrer at 100 rpm. Between the compartments, the sample was placed (23 mm diameter and 0.8 mm thickness). The test was carried out for 24 h, and at select time points of 2, 4, 6, 8, 12, and 24 h, aliquots of 100 µL solution were taken; each time, the volume was replaced with fresh simulated sweat or distilled water. Finally, the concentration of the samples was measured by absorbance, as described above. In the second trial, the penetration of active compounds in pork skin was evaluated, as explained by Schmook et al. (1993) [37]. A pig ear was obtained from a local butcher shop and the excess subcutaneous tissue was removed with a scalpel blade (fixation was not used). The remaining skin sheets were cut to a diameter of 2.3 cm, excluding sites that had scars, bruises, and earlier cuts. In this process, the skin was located with the epidermis towards the donor compartment [37]. Figure 1 illustrates the experimental setup.HistologyTo evaluate the active components’ penetration throughout the skin, pork skin tissues were complexed as described above to reveal PVI and ASA in the tissue; then, they were stained according to the protocol of Osorio et al. (2019). First, the tissue was fixed using formalin 10% *v*/*v*, and then it was dehydrated in an alcohol solution and transferred to xylene; the tissues were subsequently incorporated into paraffin. Blocks of 5 µm thickness were obtained, which were stained with hematoxylin–eosin (HE) [27]. The cuts were observed in a standard Nikon microscope equipped with a DS-Fi3 integrated camera using 10X objective magnification.Antimicrobial activityA modification of the disk-diffusion method was used to evaluate the antimicrobial activity of PVI against Gram-positive and Gram-negative bacteria. Four BNC/PVI dressings were placed on Petri dishes, where 20 mL of 0.5 McFarland microbial inoculum of *Staphylococcus aureus* and 20 mL of 0.5 McFarland microbial inoculum of *Escherichia Coli* were cultivated in 15 mL of nutritive agar and incubated for 24 h at 37 °C in a Moyco Electronics Incubator. Moreover, BNC dressings without any formulation were taken as blanks, with the aim of comparing the antibacterial effects of BNC/PVI dressings. The presence of inhibition halos at 24 h was measured and considered as an indication of antimicrobial activity against *S. aureus* and *E. coli*. Inhibition areas, subtraction of BNC/PVI diameter, average, and standard deviation calculations were performed using ImageJ software [38].Anti-inflammatory activityThe anti-inflammatory activity of BNC/ASA dressings was assessed due to the anti-inflammatory effect reported for ASA. The evaluation was performed through the method proposed by Gunathilake et al. (2018), Bashir et al. (2022), Begam et al. (2022), and Vanlalhruaii et al. (2019). A phosphate buffer solution (pH 6.4) was prepared, and 4.78 mL was mixed with 0.2 mL of the ASA formulation and 0.2 mL of bovine serum albumin (BSA) with a concentration of 50 µg/ml. The solution was placed in a water bath at 37 °C for 15 min, and then it was placed in an oven at 70 °C for 5 min. Then, the solution was cooled down to room temperature and the turbidity was measured using UV–Vis spectroscopy at 660 nm [39,40,41,42]. The denaturalization inhibition percentage is proportional to anti-inflammatory activity and was calculated according to Equation (Equation 4):
(4)$$DI\%=(1-\frac{A_{1}}{A_{2}})\cdot 100$$
where *A*${}_{1}$ is the absorbance of the control sample and *A*${}_{2}$ is the sample absorbance.Due to its natural anti-inflammatory activity, the positive control was taken as pure ASA, while the negative control was taken as a distilled water solution. Moreover, an additional sample, different from BNC/ASA, was utilized: BNC. This was used to assess the natural anti-inflammatory effect of BNC and the difference between the ASA topical formulation, the BNC/ASA dressings, BNC, and the positive control.

### 2.5. Statistical Analysis

The software ImageJ was used to measure the number of pixels and calculate the zone of inhibition in the images of the adapted standardized bacterial disk-diffusion assay. The Young’s modulus, diameters of the zones of growth inhibition (BNC/PVI) among bacteria species, and denaturalization inhibition percentage were compared per properties using one-way analysis of variance (ANOVA). The Tukey–Kramer method was performed as a post hoc test. A *p*-value < 0.05 was considered statistically significant. The statistical analysis was performed by using RStudio (R. RStudio, PBC, Boston, MA, USA).

## 3. Results

### 3.1. Development of Bioactive Wound Dressings

The appearance of the dressings is shown in Figure 2b,c. The color of the BNC/PVI dressings changed due to the presence of povidone-iodine. These dressings were produced with a diameter of 5 cm and weighed approximately 6 g in the hydrogel state. The thickness of the dressing was, on average, 0.5 mm (see Figure 2a).

### 3.2. Wound Dressing Characterization

#### 3.2.1. Scanning Electron Microscopy (SEM)

The SEM morphology of the unloaded BNC membrane is shown in Figure 3. A uniform 3D network of entangled nanoribbons that was randomly oriented was observed. The nanoribbons presented a width between 40 and 70 nm, as reported in the literature [43].

#### 3.2.2. Mechanical Characterization

The uniaxial tensile test (see Figure 4) of BNC/PVI and BNC/ASA dressings showed general viscoelastic behavior during stretching. The elastic region of the dressings represented 1.25% and 5% for BNC/PVI and BNC/ASA, respectively. The plastic region of dressings changed from 1.25% to 11% for BNC/PVI and from 5% to 21.5% for BNC/ASA. The tensile strength was, on average, 0.36 and 0.34 MPa for BNC/PVI and BNC/ASA, respectively. Table 1 summarizes the Young’s modulus, tensile strength, and elongation at break for both dressings and BNC.

According to Table 1, the BNC/PVI dressings were stiffer than BNC/ASA and presented lower elongation at break, while the inclusion of bioactive compounds (PVI and ASA) reduced the mechanical performance of BNC.

#### 3.2.3. Loading and Release Kinetics

Loading kinetics can be observed in Figure 5. For ASA and PVI, it was observed the rapid loading of the bioactive compound in the first 5 min, and it reached equilibrium after 10 min.

To model the kinetic loading behaviors, Lagergren’s pseudo-first-order model was used, which can be seen in Figure 5. The equations obtained for the modeling of each event are shown in Figure 5. $Q_{e}$ for PVI was 589.36 mg PVI/g BNC, with a K value of 5.145 min ${}^{-1}$. For ASA, $Q_{e}$ was 38.61 mg ASA/g BNC, with a K value of 0.659 min ${}^{-1}$. The above results indicate better affinity between BNC and PVI than NBC and ASA.

Regarding the release analysis, Figure 6a shows the release kinetics for BNC/PVI in water and simulated sweat media. At 24 h, the BNC/PVI dressings released 29% of PVI in water and 10% in simulated sweat. It was observed that PVI tended to remain in the BNC, rather than being released, an effect that was highlighted under simulated sweat. Release kinetics for BNC/ASA in water and simulated sweat media are shown in Figure 6b The release behavior was similar in both media, reaching a constant release rate after 3 h. BNC/ASA dressings released 29% and 26% of ASA at 24 h in water and simulated sweat, respectively.

According to Figure 6, BNC/ASA showed faster release on water, simulated sweat, and simulated sweat through pork skin than BNC/PVI, being released at approximately 25%, 22%, and 4% ASA in the first 5 h, compared to the 3%, 3%, and 0.75% released by BNC/PVI at the same time, respectively. However, although both displayed the same active compound release in water, at 24 h on pork skin, BNC/PVI yielded a greater release percentage than BNC/ASA.

#### 3.2.4. Histology

Release through the porcine dermis for the PVI and ASA is shown in Figure 7a,b and Figure 7c,d, respectively. The images show complete diffusion through the skin, which is consistent with the presence of the active compounds in the receptor chamber at the end of the experiment. As expected, a higher concentration was observed in the side facing the donator chamber and a lower one in the receptor chamber side. A considerable amount of the active compound could pass through the outer layer and penetrate more deeply into the skin after 24 h, as seen by the color gradient, which corresponds to the total amount of active compounds released by the dressings.

#### 3.2.5. Antibacterial Activity

Figure 8 and Figure 9 show the inhibition halos of BNC and BNC/PVI. BNC/PVI displayed an inhibition halo for *S. aureus* and *E. coli* of approximately 20.65 mm and 11.23 mm, respectively, after 24 h. During the evaluation, the adequate release of the active substance was perceived since PVI presented a transition from red copper to a transparent color (see Figure 8). Unloaded BNC did not present an inhibition halo; therefore, it did not possess any antibacterial activity. Bernardelli et al. (2019) reported that once the inhibition halo is observed, antibacterial activity is present in the system [44]. These results indicate that the antibacterial activity of PVI/BNC is due to the presence of PVI exclusively. Moreover, obtained data were similar to the values reported by EUCAST for the disk-diffusion method against *E. coli* and *S. aureus*. For instance, it was reported that the *E. coli* inhibition halo for samples exposed to Cefadroxil, Nitrofurantoin, Neomycin, and Netilmicin ranged from 15 to 21 mm [45], and for *S. aureus* exposed to Gentamicin, Tobramycin, and Neomycin, it ranged from 18 to 23 mm [45]. Finally, regarding the effect of the BNC/PVI dressings, the *p*-value < 0.05 indicated that the effect was different, and it was higher for *S. aureus* than *E. coli*, indicating greater activity for Gram-positive bacteria.

#### 3.2.6. Anti-Inflammatory Activity

Figure 10 shows the anti-inflammatory activity of the ASA topical formulation and BNC/ASA dressing, a positive and negative control, as well as the unloaded BNC dressing that was also investigated.

These results showed the inhibition of protein denaturation of approximately 65% for the positive control, and approximately 44% inhibition for the BNC/ASA, indicating the anti-inflammatory activity of BNC/ASA dressings. Unloaded BNC and distilled water did not display activity at all.

## 4. Discussion

SEM (Figure 2) revealed the entangled network of BNC nanoribbons that is related to the large surface area and open porosity, which facilitate the absorption of the bioactive compounds [27]. The mechanical characterization (Figure 3) revealed differences in Young’s modulus, tensile strength, and elongation at break due to the presence of PVI and ASA, molecules that can act as plasticizer agents, as reported for other biomaterials [46,47]. According to the literature, plasticizer agents can interpose within the fibrillary structure of the membrane, decreasing its rigidity and increasing its mobility and, subsequently, reducing the Young’s modulus and tensile strength of the membrane and increasing its plastic zone [46,47]. Due to the small size of the ASA molecule compared to the PVI molecule, this compound can be interposed more easily in the BNC fibrillary network and exert a greater plasticizing effect than PVI. Despite a reduction in the tensile strength (see Table 1), the materials can be manipulated. Moreover, the mechanical properties of the dressings can influence their final application in the treatment of wounds in skin [48]. In fact, the Young’s modulus in the BNC/ASA is 5.94 ± 1.34 MPa and it is 7.00 ± 1.32 MPa for BNC/PVI, which are lower values than those presented in some parts of the human body, which range from 37.66 ± 36.41 MPa to 132.47 ± 36.49 MPa, as reported by Aisling Ní Annaidh et al. (2013) [49], preventing the wound dressings from generating excessive mechanical wear.

Considering that the dressings are mechanically suitable for the desired application, it is now important to analyze the loading and release process of PVI and ASA. It was found that PVI and ASA present a positive loading constant (5.145 min${}^{-1}$ and 0.659 min${}^{-1}$, respectively), which is an indicator of the affinity between the bioactive compounds and BNC. This behavior can be explained through the molecular structure and weight of the compounds. PVI is a compound with a high molecular weight and ionic interactions between I${}^{-}$ and H${}^{+}$ in the cellulose hydroxyl groups, while the interactions between ASA and BNC occur exclusively via hydrogen bonds between hydroxyl groups from cellulose and carboxyl groups from ASA. Nevertheless, both compounds are efficiently adsorbed on BNC. The loading efficiency is approximately 95.08 ± 2.24% for PVI and 80.07 ± 1.83% for ASA. Additionally, we observed a high degree of loading for both compounds per gram of BNC, 589.36 mg for PVI and 38.61 mg for ASA, as is indicated in Figure 5.

Likewise, release assays showed differences between PVI and ASA (see Figure 6). Initially, similar release behaviors between dressings in water and dressings in simulated sweat were expected because simulated sweat’s main component is water [36]. However, simulated sweat also contains acidic (pH = 4.3) compounds, which deliver charges to the medium; these charges are further rejected by the I${}^{-}$ ions within the PVI, resulting in a slower release level under simulated sweat [15,36]. Otherwise, the acidic conditions of simulated sweat did not affect ASA delivery, explained by the molecular structure of ASA (salicylic and acetyl groups on its surface) [22,36], which easily creates hydrogen bonds with water (including under sweat conditions), resulting in the same delivery profile for both liquids [36]. The release profile obtained using porcine skin for both compounds was similar and resembled slow drug delivery across skin layers, which is advantageous for a sustained therapeutic effect.

To evaluate the bioactive components’ penetration throughout the skin, pork skin was selected due to its structural and histological similarity with human skin—for instance, the similarity in thickness and hair follicle density (20/cm${}^{2}$) [50]. In the literature, the major route described for skin permeation is through the epidermis; the pathway may include a transcellular route through the corneocytes of the stratum corneum or an intercellular route through the lipids of SC [46]. The diffusion into the intercellular lipid matrix is recognized as a determinant of drug loading by the skin, due to its large surface area [17]; it can be seen in Figure 7 as dark purple and violet (indicated by an asterisk) domains, showing that the active compounds distribute across the epidermis, into the intercellular lipid matrix and dermis. The complete permeation of PVI and ASA through all layers of the pork skin can be explained due to the nature of the compounds used in each formulation: they present a molecular weight smaller than 600 Da (ASA: 180 Da; PVI: 364 Da), which leads to a higher diffusion coefficient throughout the skin [48,51,52,53,54,55].

Further assessments were focused on the bioactivity performance of the dressings. For BNC/PVI, we found higher antibacterial activity against Gram-positive (*S. Aureus*) bacteria (see Figure 9), favoring successful wound healing since the vast majority of bacterial infections of the skin are caused by Gram-positive bacteria [45,56]. Moreover, the antimicrobial agent must be slowly released over an extended period; in this way, the active substance will fulfill its function in dressing the wound for a longer period (as indicated by the release experiments), reducing the dressing change frequency determined by the patient’s health practitioner. In our study, this behavior was observed in BNC/PVI, which presents slow active compound release after application and continues in this way for a long period (>24 h). Accordingly, BNC/PVI could have benefits in clinical practice, as dressing changes interfere with the wound healing process, removing the newly produced keratinocytes. Furthermore, the antibacterial properties of the BNC/PVI dressings show a difference among the mean diameter values of halos that can be explained by the cellular structure of the Gram-positive and Gram-negative bacteria. Gram-negative bacteria have a wall between two lipid bilayers, creating lower susceptibility in *E. coli* [57]. Nevertheless, BNC/PVI displayed acceptable bacterial inhibition for both, which was mainly due to the antiseptic action of the iodophor group, which acts by changing the cell membrane permeability, deactivating microbial enzymes by protein oxidation, and altering and deactivating the intracellular unsaturated fatty acid composition of phospholipids and DNA structural damage caused by the iodization of the pyrimidine base and amino acid derivatives [21,28,58,59].

Inflammation is an essential process during healing; however, it is important to control this in order to prevent the formation of a chronic wound and thus encourage the prompt start of the re-epithelialization process [60]. Accordingly, due to the constant release rate after 3 h and the release capacity of 29% and 26% of ASA at 24 h in water and simulated sweat, respectively, along with the significant anti-inflammatory activity (see Figure 10), BNC/ASA dressings can offer the ideal quick and continuous release within the first five hours after its application. Delivering a sufficient quantity at this time, the release occurs over several hours; in this way, the inflammatory process would be regulated in the first instance, and as the inflammation is reduced, the pain is diminished by its analgesic property, and later, the process of re-epithelialization in the wound would be resumed. Likewise, a study conducted by S. R. Bareggi et al. (1998) evidenced pain relief of up to 70% in patients with inflammatory skin conditions such as acute herpetic neuralgia and post-herpetic neuralgia during the first 5 h after an ASA topical solution was applied compared to an oral dose, inducing a higher ASA concentration in the outer skin layers [61]. Unloaded dressings and a negative control showed null inhibition on protein denaturation (anti-inflammatory effect). It has been proven that protein denaturation is related to inflammatory processes in the human body [37,62,63]. The mechanisms of the topical action of ASA are complex and somewhat unique to nonsteroidal anti-inflammatory drugs; according to Y. Mizushima et al. (1968), inhibiting protein denaturation is a mechanism of many acidic nonsteroidal anti-inflammatory drugs (NSAIDs) which interact with plasma proteins and prevent the coagulation of macromolecules; also, NSAIDs inhibit endogenous prostaglandins production by blocking COX enzyme, therefore, the inhibition of protein denaturation represents anti-inflammatory properties [40,64,65]. Overall, in the literature, the percentage of protein denaturation inhibition for plain aspirin is reported to be at least 33.61% [66]; therefore, the results indicate that the BNC/ASA dressing has anti-inflammatory activity, with an inhibition percentage of 44.83%. Thus, it was possible to develop dressings using BNC that presented specific properties (antibacterial and anti-inflammatory), which can be used in diverse types of wounds and in different cases according to the criteria of healthcare providers.

## 5. Conclusions

In this work, bacterial nanocellulose wound dressings were developed, using PVI or ASA as bioactive compounds, taking advantage of the entangled nanostructure network of BNC nanoribbons. The mechanical properties of the dressings indicate a reduction of mechanical performance due to the incorporation of the compounds, however, the dressings are still suitable for wound treatment applications. PVI had a stronger affinity for the BNC membrane and showed greater loading than ASA, due to ionic interactions between the compounds (according to loading kinetics). Moreover, the dressing displayed a slow drug delivery profile and could reach across all skin layers. BNC/PVI dressings presented antimicrobial properties for Gram-positive and Gram-negative bacteria, being higher for Gram-positive (*S. aureus*), a bacterium that invades human skin under skin and soft tissue infections. Regarding BNC/ASA, the dressings displayed anti-inflammatory properties as seen by the inhibition of protein denaturation, also, it showed a high drug delivery profile compared to BNC/PVI and the compound could reach all skin layers. These results show the beneficial properties and great potential for clinical practice of BNC dressings; therefore, this paper demonstrates that BNC is a promising platform for the development of drug delivery dressings for wound treatment applications in skin and soft tissue infections (SSTI), incorporating antimicrobial or anti-inflammatory compounds.

## Figures and Tables

**Figure 1 pharmaceutics-14-01661-f001:**
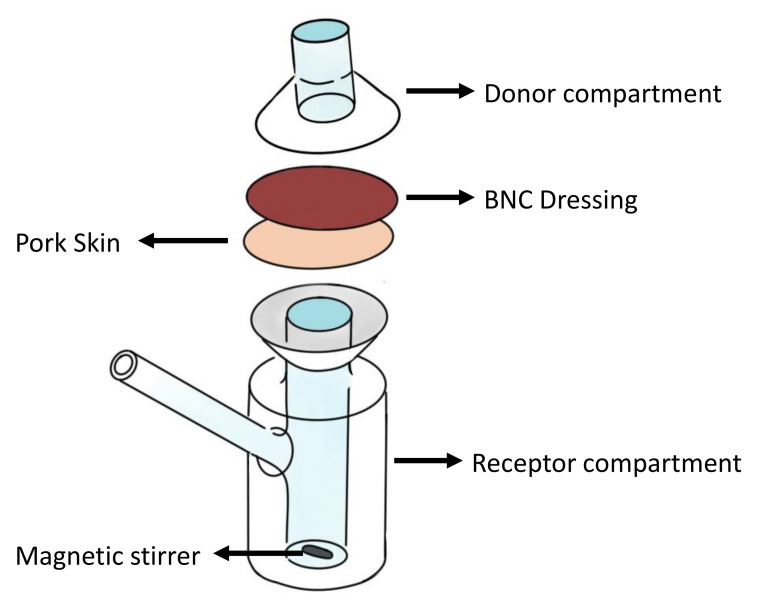
Schematic presentation of release assay.

**Figure 2 pharmaceutics-14-01661-f002:**
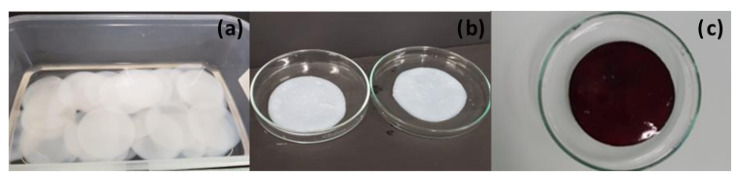
Appearance of BNC dressings—ø: 5 cm. (**a**) BNC before the incorporation of bioactive compounds. (**b**) BNC/ASA dressings, and (**c**) BNC/PVI dressings.

**Figure 3 pharmaceutics-14-01661-f003:**
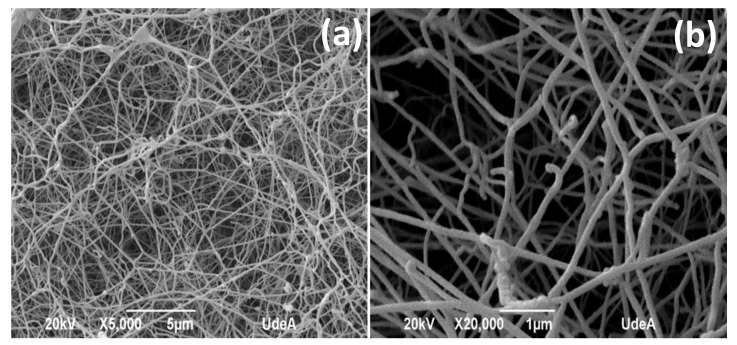
SEM images of unloaded BNC. (**a**) 5000 magnifications; (**b**) 20,000 magnifications, *n* = 1.

**Figure 4 pharmaceutics-14-01661-f004:**
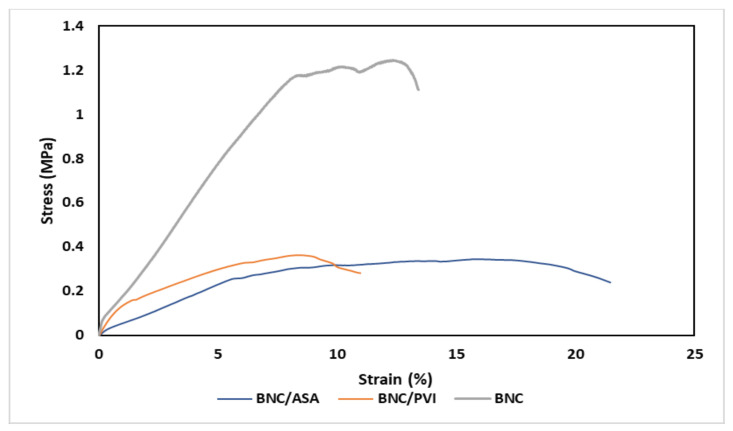
Representative stress/strain curves of BNC dressings. For this assay, 6 samples for each dressing were analyzed, along with BNC without loading (*n* = 6).

**Figure 5 pharmaceutics-14-01661-f005:**
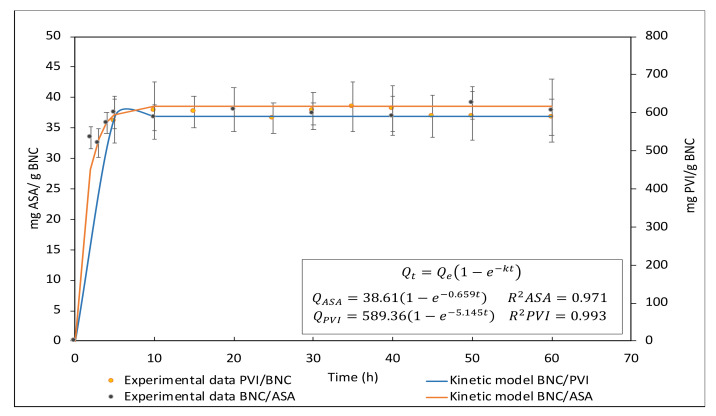
Lagergren’s pseudo-first-order model applied to the loading kinetics of bioactive compounds on BNC. Inset shows equations obtained for the modeling of each event as well as their corresponding correlation coefficient. Each assay was carried out in triplicate (*n* = 3).

**Figure 6 pharmaceutics-14-01661-f006:**
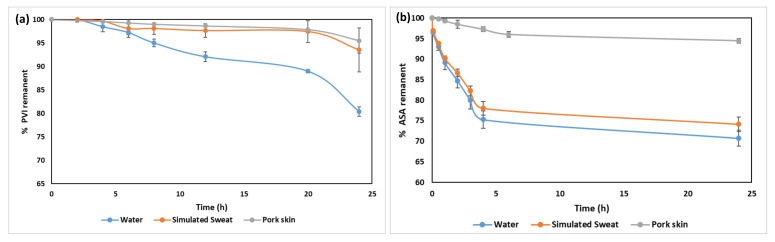
Release profiles under water, simulated sweat, and across pork skin. (**a**) Release kinetics of BNC/PVI; (**b**) release kinetics of BNC/ASA. Each assay was carried out in triplicate (*n* = 3).

**Figure 7 pharmaceutics-14-01661-f007:**
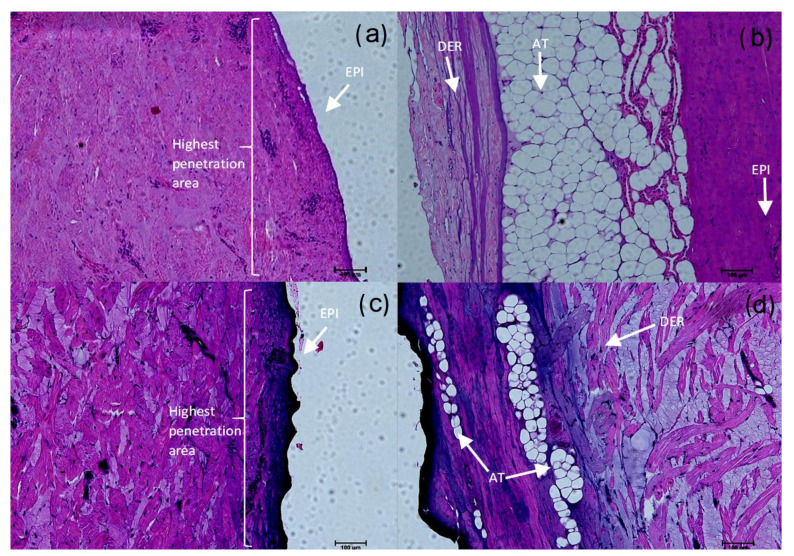
Micrographs of pork skin after Frank cell release assay of BNC/ASA and BNC/PVI dressings at 10×, in epidermis (EPI), dermis (DER), and adipose tissue (AT). (**a**) PVI in the EPI (bright violet); (**b**) PVI in the dermis (bright violet); (**c**) PVI in the EPI (dark purple); (**d**) PVI in the DER (dark purple).

**Figure 8 pharmaceutics-14-01661-f008:**
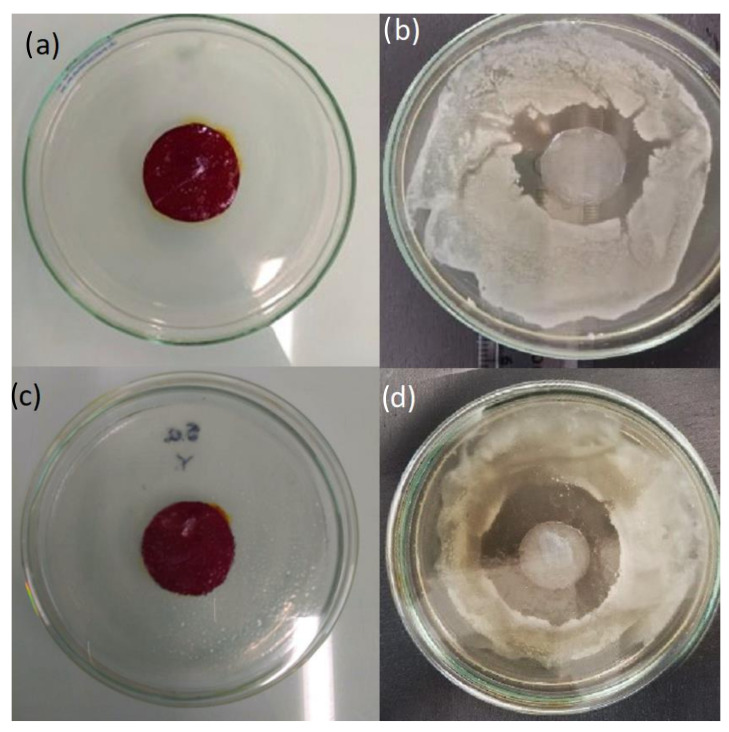
BNC/PVI dressing during the antimicrobial assay. (**a**) Immediately after placing the BNC/PVI dressing above agar for *Escherichia Coli*. (**b**) Inhibition halo after 24 h for *Escherichia Coli*. (**c**) Immediately after placing the BNC/PVI dressing above agar for *Staphylococcus aureus*. (**d**) Inhibition halo after 24 h for *Staphylococcus aureus*. Three dressing specimens were used for each strain (*n* = 3).

**Figure 9 pharmaceutics-14-01661-f009:**
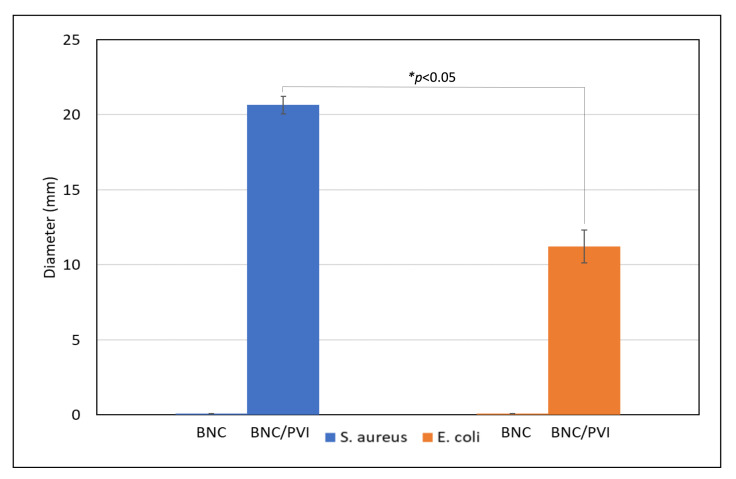
Inhibition halo diameters for BNC/PVI dressings. *n* = 3.

**Figure 10 pharmaceutics-14-01661-f010:**
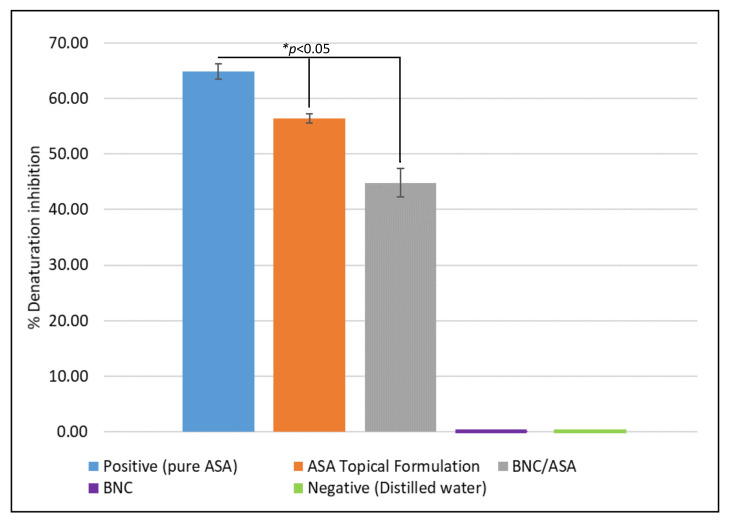
Denaturation inhibition of ABS with BNC/ASA dressing. *n* = 3.

**Table 1 pharmaceutics-14-01661-t001:** Mechanical properties of BNC dressings.

Dressing	Young’s Modulus (MPa)	Tensile Strength (MPa)	Elongation at Break (%)
BNC	28.73 ± 6.79 *	1.94 ± 0.41 *	13.63 ± 1.29 *
BNC/ASA	5.94 ± 1.34 *	0.34 ± 0.09	21.50 ± 1.76 *
BNC/PVI	7.00 ± 1.32 *	0.36 ± 0.09	11.00 ± 1.77 *

* Groups are statistically different.

## Data Availability

The data presented in this study are available on request from the corresponding author.

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
