# Peer review of "Assessment of Bacterial Nanocellulose Loaded with Acetylsalicylic Acid or Povidone-Iodine as Bioactive Dressings for Skin and Soft Tissue Infections"

_pharmaceutics, 2022, doi:10.3390/pharmaceutics14081661_

Round 1

Reviewer 1 Report

The manuscript deals with an important topic; the research project is well designed and was carried out with appropriate techniques and an impeccable execution. The manuscript is, however, weakened by an inadequate presentation of the results and has a few shortcomings that should be easily corrected.

- Figure 6 is not acceptable as it is. The figures are too small; the "bright violet" and "deep purple" are almost the same color and the situation is aggravated by the gray background. The colors and the background should be corrected and  the Figure made larger.

- The figure S1 is, in my opinion, highly relevant and should be included in the manuscript rather than confined in the Supplementary Material.

- The paper is, in general, well written but includes occasional paragraphs of uncertain construction. A final revision by a native English speaker could improve its quality and readability.

Author Response

On behalf of the authors of the paper entitle: Assessment of bacterial nanocellulose loaded with acetylsalicylic

acid or povidone iodine as bioactive dressings for skin and soft tissue infections.” Summited to the Journal Pharmaceutics, I kindly respond to the reviewers’ comments:

Comment

Response

(1) Figure 6 is not acceptable as it is. The figures are too small; the "bright violet" and "deep purple" are almost the same color, and the situation is aggravated by the gray background. The colors and the background should be corrected, and the Figure made larger.

Thank you for this suggestion, the colors and brightness of the figure 6 (Figure 7 in the corrected version) were corrected and its size was modified too.

(2) The figure S1 is, in my opinion, highly relevant and should be included in the manuscript rather than confined in the Supplementary Material. 

As suggested, we added it as the new Figure 8. The subsequent figures have been renumbered accordingly.

(3) The paper is, in general, well written but includes occasional paragraphs of uncertain construction. A final revision by a native English speaker could improve its quality and readability

Thank you for the nice reminder. The revision was made using MDPI author services.

All changes were highlighted (in green) in the main LaTEX document and the Track Changes is on. We hope all comments were accurately assessed. Please do not hesitate to contact me if further changes or comments are need.

Sincerely,

       Marlon Andrés Osorio Delgado I.Q, PhD

                                                                                 Professor of Nanotechnology Engineering

                                                                                 New Materials Research Group

                                                                                 Universidad Pontificia Bolivariana

                                                                                 Circular 1, No. 70-01, Bloque 11

                                                                                 AA 56006

                                                                                 Medellín (Colombia)

                                                                                 Fax. 57 4 4488388

Reviewer 2 Report

Although the topic of the manuscript is of interest and relevance, the authors should address some points. 

·       The quality of the scientific writing and English language should be improved and revised. I would recommend having the manuscript revised by a native speaker.

·       Figure 1: If possible, a schematic image representing the process of BNC production/isolation would be helpful.

·       Figure 2: it would be interesting to see SEM images of ASA/PVI loaded membranes to evaluate possible changes in fibre morphology/diameter.

·       Figure 3: please add mechanical measurements of unloaded BNC membrane.

·       Figure 6: If possible, a schematic image representing the assay performed would be helpful. Please separate the 4 images.

Author Response

On behalf of the authors of the paper entitle: Assessment of bacterial nanocellulose loaded with acetylsalicylic

acid or povidone iodine as bioactive dressings for skin and soft tissue infections.” Summited to the Journal Pharmaceutics, I kindly respond to the reviewers’ comments:

Reviewer 2

Comment

Response

(1) The quality of the scientific writing and English language should be improved and revised. I would recommend having the manuscript revised by a native speaker.

Thank you for the comment. The revision was made using MDPI author services.

(2) Figure 2 it would be interesting to see SEM images of ASA/PVI loaded membranes to evaluate possible changes in the fiber morphology.

Thank you very much for pointing this out, however, it is not possible because the sample is a hydrogel as such is hydrated and for a SEM analysis the sample should be dry. If the hydrated sample is analyzed through SEM, we would see the liquid surface and not the material. Unfortunately, bioactive components are liquids that does not dry using freeze drying, thus only the unloaded membrane could be dried as showed in figure 3.

(3) Figure 3: please add mechanical measurements of unloaded BNC membrane.

Added as requested.

(4) Figure 6: If possible, a schematic image representing the assay performed would be helpful. Please separate the 4 images

As suggested, we added a schematic image of the desorption assay. See NEW Figure 1.

All changes were highlighted (in green) in the main LaTEX document and the Track Changes is on. We hope all comments were accurately assessed. Please do not hesitate to contact me if further changes or comments are need.

Sincerely,

       Marlon Andrés Osorio Delgado I.Q, PhD

                                                                                 Professor of Nanotechnology Engineering

                                                                                 New Materials Research Group

                                                                                 Universidad Pontificia Bolivariana

                                                                                 Circular 1, No. 70-01, Bloque 11

                                                                                 AA 56006

                                                                                 Medellín (Colombia)

                                                                                 Fax. 57 4 4488388

Reviewer 3 Report

Dear Authors,

Your manuscript is generally well-written and interesting, nevertheless, there are a few things which should really be improved:

Major remarks:

1.Title: it is misleading. From what I understand You tested iodine OR asa- containing dressings; the word "AND" suggest that specific dressing was immersed in both bioactive substances. By the way, it would be very interesting to check how iodine influences asa performance.

2. Antimicrobial activity: it is a total mess now, but You should be able to improve it quickly. Firstly - did You perform modification of disk-diffusion method (0.5McFarland's inoculum swabbed?) it can not be understood from the description in MM. If not - why? Make Your MM more specific and transparent. Secondly, how did You measure the halo zones diameters ( did it include the size of BC or You substracted the size of BC - this is important). To make it more transparent, I also suggest to add the data on BC diameter in Figure 1 caption.

Discussion: Why have You chosen specifically iodine? Please discuss, there are many relevant papers on the topic, please refer for example to: Dydak et al: 2021 In vitro Efficacy of Bacterial Cellulose Dressings Chemisorbed with Antiseptics Against Biofilm Formed by Pathogens Isolated from Chronic Wounds.

Minor remarks:

please check the spelling carefully throughout the manuscript - sometimes the errors are startling ("young modulus" instead of "Young modulus").

did You use PVI or PVP-I?

Author Response

On behalf of the authors of the paper entitle: Assessment of bacterial nanocellulose loaded with acetylsalicylic

acid or povidone iodine as bioactive dressings for skin and soft tissue infections.” Summited to the Journal Pharmaceutics, I kindly respond to the reviewers’ comments:

Reviewer 3

Comment

Response

(1) Title: it is misleading. From what I understand you tested iodine OR asa- containing dressings; the word "AND" suggest that specific dressing was immersed in both bioactive substances. By the way, it would be very interesting to check how iodine influences asa performance.

Thanks for the comment. The title was corrected and the possible influence of povidone iodine over ASA performance would be an exciting potential research topic for the future.

(2) Antimicrobial activity: it is a total mess now, but You should be able to improve it quickly. Firstly - did You perform modification of disk-diffusion method (0.5McFarland's inoculum swabbed?) it cannot be understood from the description in MM. If not - why? Make Your MM more specific and transparent. Secondly, how did You measure the halo zones diameters (did it include the size of BC, or You subtracted the size of BC - this is important). To make it more transparent, I also suggest adding the data on BC diameter in Figure 1 caption

Thank you very much for the questions. Indeed, we perform a modification of disk-diffusion method. And we subtracted the diameter of BNC to obtain the real measure of the inhibition halo zones. This information was added to MM (see line 188), and the sentences were revised.

(3) Why have You chosen specifically iodine? Please discuss, there are many relevant papers on the topic, please refer for example to: Dydak et al: 2021 In vitro Efficacy of Bacterial Cellulose Dressings Chemisorbed with Antiseptics Against Biofilm Formed by Pathogens Isolated from Chronic Wounds.

Thanks for the kind suggestion. A greater explanation of PVI mechanism of action and relevance is now included in the Introduction (see line 137), and in the Discussion (see line 331). Also, the new reference was added when needed.

Minor: please check the spelling carefully throughout the manuscript - sometimes the errors are startling ("young modulus" instead of "Young modulus").

Revised as requested.

Minor: did You use PVI or PVP-I?

PVI is one of many abbreviations for Povidone Iodine along with PI and PVP-I, even though the abbreviations are different it is still the same compound. Some examples are: Piotr Kanclerz et al [1]  use PVI as abbreviation, meanwhile Firas Ayoub et al [2] use PI as abbreviation, and Daniela Vergara et al [3]use PVP-I, all of them referring to the same compound Povidone Iodine.

All changes were highlighted (in green) in the main LaTEX document and the Track Changes is on. We hope all comments were accurately assessed. Please do not hesitate to contact me if further changes or comments are need.

Sincerely,

       Marlon Andrés Osorio Delgado I.Q, PhD

                                                                                 Professor of Nanotechnology Engineering

                                                                                 New Materials Research Group

                                                                                 Universidad Pontificia Bolivariana

                                                                                 Circular 1, No. 70-01, Bloque 11

                                                                                 AA 56006

                                                                                 Medellín (Colombia)

                                                                                 Fax. 57 4 4488388

References:

[1]      P. Kanclerz and W. G. Myers, “Chlorhexidine and other alternatives for povidone-iodine in ophthalmic surgery: review of comparative studies,” J Cataract Refract Surg, vol. 48, no. 3, pp. 363–369, Mar. 2022, doi: 10.1097/J.JCRS.0000000000000754.

[2]      F. Ayoub, M. Quirke, R. Conroy, and A. Hill, “Chlorhexidine-alcohol versus povidone-iodine for pre-operative skin preparation: A systematic review and meta-analysis,” International Journal of Surgery Open, vol. 1, pp. 41–46, Jan. 2015, doi: 10.1016/J.IJSO.2016.02.002.

[3]      D. Vergara, N. Loza-Rodríguez, F. Acevedo, M. Bustamante, and O. López, “Povidone-iodine loaded bigels: Characterization and effect as a hand antiseptic agent,” Journal of Drug Delivery Science and Technology, vol. 72, p. 103427, Jun. 2022, doi: 10.1016/J.JDDST.2022.103427.

Reviewer 4 Report

Argel et al describe a nanocellulose scaffold impregnated with povidone iodine to reduce infection risk and acetylsalicylic acid (aka aspirin) to modulate inflammation when implanted in an injury site. Material/drug combination products are an important next phase of medical device development, with direct clinical applications in wound care. The authors provide an in vitro and ex vivo characterization of the wound dressing, including mechanics, structure, and drug release. There are some questions about methodology and experimental interpretation, and overall the impact of the study would be greatly increased with in vivo studies of efficacy in an infection/wound model.

Abstract/Introduction

1.     Do you have any references for PVI cell compatibility? Would this compound be appropriate for implantation (i.e. soft tissue repair) or only for topical wound treatment?

2.     Though it may be true, what literature supports the statement that “PVI has been the most widely investigated for use in wound dressings?” Does it mean that the most studies or products have used? If so, the provided reference #5 doesn’t state that.

Methods/Results

3.     What are the dimensions of the dressings when synthesized? e.g. what is the thickness?

4.     The adsorption/desorption experiments were not clear as described. What material sizes, volumes, etc were used? Was there agitation? Furthermore, if the wound dressing is initially hydrated, is adsorption analysis possible? Simple dilution may be occurring as PVI solution diffuses into the hydrated dressing. Similarly for desorption, there is likely substantial free liquid prior to drying that contains the compound of interest. So rather than being adsorbed it is very possible that it’s drying as a film. This may be addressed by rinsing excess compound after lyophilizing the dressing. Under these testing conditions absorption may be playing a prominent role.

5.     Tissue diffusion is an excellent idea for an experiment. Was the pig tissue fixed prior to processing and does the complex remain trapped in the tissue throughout tissue processing?

6.     The anti-inflammatory analysis is unexpected and it’s unclear how we should interpret the assay results. The known inflammation modifying mechanisms of ASA (an NSAID) are mediated through cellular pathways, most notably (but not exclusively) as a COX inhibitor. I’m not aware of BSA denaturation under high heat as a valid assay of immune modulation, nor can I envision a hypothetical mechanism of this. Reference 34 supporting “It has been proven that protein denaturation is related to inflammatory processes in the human body [34].” is problematic as this reference does not mention denaturation nor inflammation. Though there is an association of protein misfolding with various pathologies, what references support the assertion that ASA modifies inflammation via this mechanism in vivo? Substantial evidence from the literature is necessary or this cannot be described as an anti-inflammatory assay.

7.     Every figure should include n-value. (how many test articles, from how many batches)

8.     Does “the BNC properties were taken from Chen et al.(2018)” mean the data points came from another study? A direct statistical comparison to groups using a different testing set up may not be appropriate.

9.     Figure 6: A different stain other than H&E will be very helpful if the complex also stains purple. Purple staining is common with hematoxylin/eosin making it very difficult to see specific signal here. (I also suggest increasing the brightness for the H&E). A suggestion is to use a different counterstain or only a very lightly stained hematoxylin only to give more contrast to the purple staining drugs. Serial sections of the H&E are still highly valuable for reference and should be kept in the figure.   

10.  Figure 7: please use a consistent group order (BNC always first for example)

11.  Is the statement “with p-value<0.05, means that the effect was different, and higher on S. aureus than E. coli, indicating greater activity for gram-positives bacteria.” supported by the data? Both groups appear to be essentially 0 on this scale though the different strains have greater starting burden. If the authors would like to make this statement, please perform a statistical comparison.

Author Response

On behalf of the authors of the paper entitle: Assessment of bacterial nanocellulose loaded with acetylsalicylic

acid or povidone iodine as bioactive dressings for skin and soft tissue infections.” Summited to the Journal Pharmaceutics, I kindly respond to the reviewers’ comments:

Reviewer 4

Comment

Response

(1) Do you have any references for PVI cell compatibility? Would this compound be appropriate for implantation (i.e., soft tissue repair) or only for topical wound treatment?

Thanks for the comment. The aim of the paper was to evaluate Bacterial Cellulose as a topical wound dressing and not be used as an implantable biomaterial; therefore, we did not present any cell compatibility for PVI or related information. Moreover, PVI is a bioactive compound widely used in treating infected wounds in clinic.

(2) Though it may be true, what literature supports the statement that “PVI has been the most widely investigated for use in wound dressings?” Does it mean that the most studies or products have used? If so, the provided reference #5 doesn’t state that.

Thank you for the question. We revised the sentence as follows: “PVI has been one of the most widely used antiseptic on skin” (see line 137) and a new reference was provided.

(3) What are the dimensions of the dressings when synthesized? e.g., what is the thickness?

It is a good question. Bacterial cellulose can grow in any kind of dimensions, which can be controlled by modifying the days of incubation and the template the culture media is pour over. The specific dimensions used here, were described in line 208.

(4) The adsorption/desorption experiments were not clear as described. What material sizes, volumes, etc. were used? Was there agitation? Furthermore, if the wound dressing is initially hydrated, is adsorption analysis possible? Simple dilution may be occurring as PVI solution diffuses into the hydrated dressing. Similarly for desorption, there is likely substantial free liquid prior to drying that contains the compound of interest. So rather than being adsorbed it is very possible that it’s drying as a film. This may be addressed by rinsing excess compound after lyophilizing the dressing. Under these testing conditions absorption may be playing a prominent role

All the adsorption experiments in hydrogel state indicates that BNC adsorbed the bioactive compounds, the oven dried was used only to remove the excess of free liquid not adsorbed in the dressing to avoid interferences in the desorption experiment. Figure 2 shows the appearance of the developed dressings where a complete integration of the compounds is present, no separation of the dressing or films of bioactive compounds were addressed.

All the requested information was added at materials and methods

(5) Tissue diffusion is an excellent idea for an experiment. Was the pig tissue fixed prior to processing and does the complex remain trapped in the tissue throughout tissue processing?

Thank you for the nice reminder. Porcine tissue was use directly without further modification to prevent morphological changes prior the desorption assay, and once it was carried out, the skin was fixed with 10% formaldehyde for subsequent evaluation. A proper explanation was added in the experimental section

(6) The anti-inflammatory analysis is unexpected and it’s unclear how we should interpret the assay results. The known inflammation modifying mechanisms of ASA (an NSAID) are mediated through cellular pathways, most notably (but not exclusively) as a COX inhibitor. I’m not aware of BSA denaturation under high heat as a valid assay of immune modulation, nor can I envision a hypothetical mechanism of this. Reference 34 supporting “It has been proven that protein denaturation is related to inflammatory processes in the human body [34].” is problematic as this reference does not mention denaturation nor inflammation. Though there is an association of protein misfolding with various pathologies, what references support the assertion that ASA modifies inflammation via this mechanism in vivo? Substantial evidence from the literature is necessary or this cannot be described as an anti-inflammatory assay

Thank you very much for pointing this out. A couple of new references were added to explain that protein denaturation is related to inflammatory processes, and it has been an assay previously used by different authors to evaluate the anti-inflammatory activity of compounds. E.g. Bashir et al[1] , Begam et al[2], Vanlalhruaii et al[3] .

(7) Every figure should include n-value. (How many test articles, from how many batches)

Revised as requested.

(8) Does “the BNC properties were taken from Chen et al. (2018)” mean the data points came from another study? A direct statistical comparison to groups using a different testing set up may not be appropriate.

We agree, the assay was performed for BNC under the same conditions and new data points are presented. The correction was made.

(9) Figure 6: A different stain other than H&E will be very helpful if the complex also stains purple. Purple staining is common with hematoxylin/eosin making it very difficult to see specific signal here. (I also suggest increasing the brightness for the H&E). A suggestion is to use a different counterstain or only a very lightly stained hematoxylin only to give more contrast to the purple staining drugs. Serial sections of the H&E are still highly valuable for reference and should be kept in the figure

Thank you for the suggestion, the colors and brightness of the figure 6 were corrected.

(10) Figure 7: please use a consistent group order (BNC always first for example)

Revised as requested.

(11) Is the statement “with p-value<0.05, means that the effect was different, and higher on S. aureus than E. coli, indicating greater activity for gram-positives bacteria.” supported by the data? Both groups appear to be essentially 0 on this scale though the different strains have greater starting burden. If the authors would like to make this statement, please perform a statistical comparison.

Thank you for the comment, you are right a statistical analysis between strains were performed to support the statements in the analysis.

All changes were highlighted (in green) in the main LaTEX document and the Track Changes is on. We hope all comments were accurately assessed. Please do not hesitate to contact me if further changes or comments are need.

Sincerely,

       Marlon Andrés Osorio Delgado I.Q, PhD

                                                                                 Professor of Nanotechnology Engineering

                                                                                 New Materials Research Group

                                                                                 Universidad Pontificia Bolivariana

                                                                                 Circular 1, No. 70-01, Bloque 11

                                                                                 AA 56006

                                                                                 Medellín (Colombia)

                                                                                 Fax. 57 4 4488388

References:

[1]      B. Lawal et al., “Preclinical anti-inflammatory and antioxidant effects of Azanza garckeana in STZ-induced glycemic-impaired rats, and pharmacoinformatics of it major phytoconstituents,” Biomedicine & Pharmacotherapy, vol. 152, p. 113196, Aug. 2022, doi: 10.1016/J.BIOPHA.2022.113196.

[2]      R. Begam, A. Shajahan, B. Shefin, and V. Murugan, “Synthesis of novel naphthalimide tethered 1,2,3-triazoles: In vitro biological evaluation and docking study of anti-inflammatory inhibitors,” Journal of Molecular Structure, vol. 1254, p. 132364, Apr. 2022, doi: 10.1016/J.MOLSTRUC.2022.132364.

[3]      C. Malsawmtluangi and H. Lalhlenmawia, “Evaluation of in vitro anti-inflammatory activity of the spadix of Colocasia affinis,” 2019, doi: 10.33493/scivis.19.02.06.

Round 2

Reviewer 2 Report

-

Author Response

Thank you for you suggestion. The document was revised again in all english spelling.

Reviewer 4 Report

The authors have successfully addressed many of the questions and comments from the first version. These edits clarified many points and have improved the manuscript. A couple questions remain and these points should be addressed or clarified.  

·       Adsorption assay: it’s still not clear to me that this assay demonstrates that these compounds were adsorbed to the BNC material. It shows overall “loading” of the drug into the dressing but cannot distinguish adsorption specifically. As such, it is likely not appropriate to refer to it as an adsorption experiment. This doesn’t detract from the application goals, it simply clarifies that the assay characterizes total drug loading rather than a specific mechanism thereof. Loading, infusion, etc are possible alternatives, and “release” an example alternative for desorption. Adsorption describes a specific phenomenon: the accumulation and concentration of molecules at the material interface. The issue at hand is that the methods state the BNC dressing is already hydrated upon fabrication: “Dressings were purified in 5 wt. % 10 KOH solution at room temperature for 14 h, followed by continuous rising with water until a neutral pH was reached.” Thus, the BNC delivers additional solvent volume to the system which will alter concentration Ct in the absence of adsorption (or even the material for that matter). Equation 1 does not account for this volume, and likely assumes dry material. If a highly accurate analysis of initial water content was performed for the BNC it would theoretically be possible to calculate adsorption at steady state. Likewise, if the BNC was fully dried prior to immersion in drug solution then this analysis would be appropriate.

·       Desorption assay: likewise, this assay does not describe desorption specifically, but rather more generally release and transport of compound from the BNC dressing and through skin. This language should be clarified accordingly. After drying, total loaded drug will be reconstituted with the donor fluid including both free drug that was previously in solution and adsorbed.

·       Thank you for the references for the protein denaturation assay in the response to reviewers. Please double check that these were added to the manuscript (I was unable to find them). My final request is to please add a reference showing that inhibiting protein denaturation is a mechanism of the anti-inflammatory activity for this compound or NSAIDs generally. (such as in vivo or in vitro leukocyte assays). The additional references provided describe the assay but do not seem to provide evidence that this is a contributing mechanism in addition to or independent of COX inhibition. Protein denaturation is correlated with inflammation, but this alone does not infer causation. Thus targeting denaturation would not be anti-inflammatory without evidence to that effect. Alternatively, consider rephrasing more generally such as “activity” with speculation on anti-inflammatory properties in the discussion. (Or a cell based assay would be the most effective means of showing activity).

Author Response

Medellín, August 1, 2022

On behalf of the authors of the paper entitle: Assessment of bacterial nanocellulose loaded with acetylsalicylic

acid or povidone iodine as bioactive dressings for skin and soft tissue infections.” Summited to the Journal Pharmaceutics, I kindly respond to the reviewers’ comments:

Reviewer 4

Comment

Response

It’s still not clear to me that this assay demonstrates that these compounds were adsorbed to the BNC material. It shows overall “loading” of the drug into the dressing but cannot distinguish adsorption specifically. As such, it is likely not appropriate to refer to it as an adsorption experiment. This doesn’t detract from the application goals; it simply clarifies that the assay characterizes total drug loading rather than a specific mechanism thereof. Loading, infusion, etc are possible alternatives, and “release” an example alternative for desorption. Adsorption describes a specific phenomenon: the accumulation and concentration of molecules at the material interface. The issue at hand is that the methods state the BNC dressing is already hydrated upon fabrication: “Dressings were purified in 5 wt. % 10 KOH solution at room temperature for 14 h, followed by continuous rising with water until a neutral pH was reached.” Thus, the BNC delivers additional solvent volume to the system which will alter concentration Ct in the absence of adsorption (or even the material for that matter). Equation 1 does not account for this volume, and likely assumes dry material. If a highly accurate analysis of initial water content was performed for the BNC it would theoretically be possible to calculate adsorption at steady state. Likewise, if the BNC was fully dried prior to immersion in drug solution then this analysis would be appropriate

Thank you for the suggestion, the word adsorption was substituted by loading since it was more accurate to describe the assay and research performed.

This assay does not describe desorption specifically, but rather more generally release and transport of compound from the BNC dressing and through skin. This language should be clarified accordingly. After drying, total loaded drug will be reconstituted with the donor fluid including both free drug that was previously in solution and adsorbed

Thank you for the suggestion. It was revised as requested.

Please double check that these were added to the manuscript (I was unable to find them). My final request is to please add a reference showing that inhibiting protein denaturation is a mechanism of the anti-inflammatory activity for this compound or NSAIDs generally. (such as in vivo or in vitro leukocyte assays). The additional references provided describe the assay but do not seem to provide evidence that this is a contributing mechanism in addition to or independent of COX inhibition. Protein denaturation is correlated with inflammation, but this alone does not infer causation. Thus, targeting denaturation would not be anti-inflammatory without evidence to that effect. Alternatively, consider rephrasing more generally such as “activity” with speculation on anti-inflammatory properties in the discussion. (Or a cell-based assay would be the most effective means of showing activity).

Thank you very much for the comment. The references reported in the response were added to the paper to support the response in the method section (see line 198). Additionally, as requested, a new reference was added showing that inhibiting protein denaturation is a mechanism of conventional NSAIDs, along with the capacity of inhibiting endogenous prostaglandins production by blocking COX enzyme; in this way, the anti-denaturation assay is an adequate method for evaluating anti-inflammatory properties of ASA [1].

Also, additional references were added in the introduction (see line 137) and methods (see line 198) to broad the supportive literature.

English language and style are fine/minor spell check required

The English was reviewed once more.

All changes were highlighted in the pdf. We hope all comments were accurately assessed. Please do not hesitate to contact me if further changes or comments are need.

Sincerely,

       Marlon Andrés Osorio Delgado I.Q, PhD

                                                                                 Professor of Nanotechnology Engineering

                                                                                 New Materials Research Group

                                                                                 Universidad Pontificia Bolivariana

                                                                                 Circular 1, No. 70-01, Bloque 11

                                                                                 AA 56006

                                                                                 Medellín (Colombia)

                                                                                 Fax. 57 4 4488388

References:

[1] Yesmin et al., “Membrane stabilization as a mechanism of the anti-inflammatory activity of ethanolic root extract of Choi (Piper chaba),” Clinical Phytoscience 2020 6:1, vol. 6, no. 1, pp. 1–10, Aug. 2020, doi: 10.1186/S40816-020-00207-7.